# Nutritional Mushroom Treatment in Meniere’s Disease with *Coriolus versicolor*: A Rationale for Therapeutic Intervention in Neuroinflammation and Antineurodegeneration

**DOI:** 10.3390/ijms21010284

**Published:** 2019-12-31

**Authors:** Maria Scuto, Paola Di Mauro, Maria Laura Ontario, Chiara Amato, Sergio Modafferi, Domenico Ciavardelli, Angela Trovato Salinaro, Luigi Maiolino, Vittorio Calabrese

**Affiliations:** 1Department of Biomedical and Biotechnological Sciences, University of Catania, Torre Biologica. Via Santa Sofia, 97, 95123 Catania, Italy; mary-amir@hotmail.it (M.S.); marialaura.ontario@ontariosrl.it (M.L.O.); sergio.modafferi@gmil.com (S.M.); trovato@unict.it (A.T.S.); 2Department of Medical and Surgery Sciences, University of Catania, Via Santa Sofia 78, 95123 Catania, Italy; paola_mp86@hotmail.it (P.D.M.); chiaraamato@hotmail.it (C.A.); calabres@unict.it (V.C.); 3School of Human and Scocial Science, “Kore” University of Enna, Via Salvatore Mazza 1, 94100 Enna, Italy; domenico.ciavardelli@unikore.it; 4Centro Scienze dell’Invecchiamento e Medicina Traslazionale-CeSI-Met, via Luigi Polacchi 11, 66100 Chieti, Italy

**Keywords:** redoxomics, glutathione, meniere’s disease, neurodegenerative diseases

## Abstract

Meniere’s disease (MD) represents a clinical syndrome characterized by episodes of spontaneous vertigo, associated with fluctuating, low to medium frequencies sensorineural hearing loss (SNHL), tinnitus, and aural fullness affecting one or both ears. To date, the cause of MD remains substantially unknown, despite increasing evidence suggesting that oxidative stress and neuroinflammation may be central to the development of endolymphatic hydrops and consequent otholitic degeneration and displacement in the reuniting duct, thus originating the otolithic crisis from vestibular otolithic organs utricle or saccule. As a starting point to withstand pathological consequences, cellular pathways conferring protection against oxidative stress, such as vitagenes, are also induced, but at a level not sufficient to prevent full neuroprotection, which can be reinforced by exogenous nutritional approaches. One emerging strategy is supplementation with mushrooms. Mushroom preparations, used in traditional medicine for thousands of years, are endowed with various biological actions, including antioxidant, immunostimulatory, hepatoprotective, anticancer, as well as antiviral effects. For example, therapeutic polysaccharopeptides obtained from *Coriolus versicolor* are commercially well established. In this study, we examined the hypothesis that neurotoxic insult represents a critical primary mediator operating in MD pathogenesis, reflected by quantitative increases of markers of oxidative stress and cellular stress response in the peripheral blood of MD patients. We evaluated systemic oxidative stress and cellular stress response in MD patients in the absence and in the presence of treatment with a biomass preparation from *Coriolus*. Systemic oxidative stress was estimated by measuring, in plasma, protein carbonyls, hydroxynonenals (HNE), and ultraweak luminescence, as well as by lipidomics analysis of active biolipids, such as lipoxin A4 and F2-isoprostanes, whereas in lymphocytes we determined heat shock proteins 70 (Hsp72), heme oxygenase-1 (HO-1), thioredoxin (Trx), and γ-GC liase to evaluate the systemic cellular stress response. Increased levels of carbonyls, HNE, luminescence, and F2-isoprostanes were found in MD patients with respect to the MD plus *Coriolus*-treated group. This was paralleled by a significant (*p* < 0.01) induction, after *Coriolus* treatment, of vitagenes such as HO-1, Hsp70, Trx, sirtuin-1, and γ-GC liase in lymphocyte and by a significant (*p* < 0.05) increase in the plasma ratio-reduced glutathione (GSH) vs. oxidized glutathione (GSSG). In conclusion, patients affected by MD are under conditions of systemic oxidative stress, and the induction of vitagenes after mushroom supplementation indicates a maintained response to counteract intracellular pro-oxidant status. The present study also highlights the importance of investigating MD as a convenient model of cochlear neurodegenerative disease. Thus, searching innovative and more potent inducers of the vitagene system can allow the development of pharmacological strategies capable of enhancing the intrinsic reserve of vulnerable neurons, such as ganglion cells to maximize antidegenerative stress responses and thus providing neuroprotection.

## 1. Introduction

Prosper Meniere more than 150 ago first described the disease named after him, and to date, although many studies have tried to describe the etiology of Meniere’s disease (MD), it still represents a matter of scientific debate. [1]. Among the theories considered to explain its pathophysiology, endolymphatic hydrops with disturbed longitudinal endolymph flow is considered central to MD pathology [2], widely recognized as primary cause leading to cochlear degeneration [3]. Anatomical variation in the position or size of sac and duct in the endolymphatic system, and the presence of viral, autoimmune inflammatory or genetic components are all possible contributory factors to the endolymph homeostasis [4]. Recent studies indicate a pattern of similarity between MD and benign paroxysmal positional vertigo, including age of onset, raising the conceivable possibility that detached saccular otoconia, an event which could be promoted by metabolic disturbances associated with oxidative stress, might represent the fundamental cause of MD [5]. As an hydropic ear pathology, MD is characterized by a triad of symptoms, such as episodic vertigo and tinnitus associated with fluctuating hearing loss, and endolymphatic hydrops, as found on post-mortem examination [2]. However emerging evidence has given rise to the conceivable possibility that MD is a systemic oxidant disorder, where excessive production of free radicals and oxidative stress promote microvascular damage, which is involved in the development of endolymphatic hydrops. Consequently, cellular damage and apoptotic cell death-induced cochleovestibular dysfunction ensues with significant reductions in dendritic innervation densities, and ultrastructural abnormalities reflecting the primary neurotoxic insult [6,7].

While reactive oxygen species at a physiological level play an important role in cellular signaling, excess in free-radical species or oxidative stress due to decreased expression and activity of antioxidant proteins becomes a toxic cause of accelerated aging [8,9,10]. Thus, the cellular capacity to counteract stressful conditions, known as cellular stress response, requires the activation of pro-survival pathways endowed with increased antioxidant, anti-inflammatory, and antiapoptotic potential [11,12,13]. 

Consistent with this notion, integrated survival responses exist in the central and peripheral nervous system, which are controlled by redox-dependent genes, termed vitagenes [14,15]. These include gene coding for proteins that actively operate in detecting and controlling diverse forms of stress and neuronal injuries, such as heat shock proteins (Hsps), γ-GC liase, thioredoxin, sirtuins, and Lipoxin A4 [16]. As a metabolic product of arachidonic acid, LXA4 is an endogenous ‘‘stop signal’’ for inflammatory processes, exhibiting its potent anti-inflammatory potential in various inflammatory disorders, such as arthritis, periodontitis, nephritis, or inflammatory bowel disease [17,18]. Chronic inflammation is known to be central to the progression of Alzheimer’s disease (AD), although identification of mechanisms capable of restoring an anti-inflammatory environment compromised in AD pathology remains an area of active investigation [19,20]. Treatment with the pro-resolving mediator aspirin-triggered lipoxin A4 (ATL) resulted in improved cognition, reduced Aβ levels, and enhanced microglia phagocytic activity in Tg2576 transgenic AD mice [21]. Furthermore, LXA4 levels are reduced with age, a pattern significantly more impacted in 3xTg-AD mice [22]. Moreover, in 3xTg-AD mice, up-regulation of lipoxin A4 was induced by aspirin-enhanced cognitive performance while reducing Aβ and phosphorylated-tau (*p*-tau) levels, an effect associated with astrocyte and microglia reactivity [18]. LXA4 action is mediated by LXA4 receptor (ALX) on the cellular membrane, which is known as formyl-peptide receptor-like 1 (FPRL1) [23], and activation of LXA4 signaling can well serve as a robust therapeutic target for mitigating AD-related inflammation and consequential cognitive dysfunction. 

Vitagene cellular stress response confers a cytoprotective state not only during aging but also in a variety of human diseases, including cancer, inflammation, and neurodegenerative disorders [24]. Given the broad cytoprotective potential of vitagenes there is now increasing interest in discovering and developing pharmacological agents able to induce stress responses [25]. When appropriately activated, cellular stress response restores redox equilibrium by activating antioxidant and anti-inflammatory pathways, which is of particular importance for brain cells with relatively weak endogenous antioxidant defenses, such as spiral ganglion neurons, centrally involved in the pathogenesis of MD and a preferential site for accumulation of lipoperoxidative hydroxynonenals and protein oxidation carbonyls product, which can disrupt redox homeostasis [26]. 

Mushrooms, which have been used in traditional medicine for thousands of years [27,28], are emerging as an important nutritional component in the diet capable of modulating the immunity system and inflammatory status. In Asian countries, for instance, modern clinical practice continues to rely on mushroom-derived preparations. According to this, many controlled studies have investigated a long list of mushroom extracts, showing various immunomodulatory biological actions, associated with antioxidant, antiviral, anticancer, and hepatoprotective activities [29,30]. As a result, many traditionally employed mushrooms, including extracts of *Agaricus campestris*, *Pleurotus ostreatus* and *Coriolus versicolor* have shown medicinal effects [31]. In particular, the active principle from *Coriolus versicolor* represents a new class of elements termed biological response modifiers (BRM) [32], which characterize several agents capable of stimulating the immune system, therefore exhibiting various therapeutic effects. Consistent with the neuroinflammatory pathogenesis of neurodegenerative damage occurring in AD, a recent study from our laboratory has provided convincing experimental evidence into the neuroprotective role of *Coriolus* biomass preparation against the neuroinflammatory process, evaluating also the impact of this nutritional intervention on cellular stress response mechanism operating in the central nervous system [33,34]. 

In the present study we examined the hypothesis that neurotoxic insult represents a critical primary mediator operating in MD pathogenesis, reflected by quantitative increases of markers of oxidative stress and cellular stress response in the peripheral blood of MD patients. We also explore the hypothesis that changes in lipidomics, as well as redox glutathione status associated with increased expression of neuroprotective vitagenes induced through supplementation with mushrooms biomass preparation from Mycology Research Laboratories Ltd., Luton, UK, *Coriolus v.* can provide a novel target for innovative therapeutic approaches aimed at minimizing oxidative stress, neuroinflammation, and neurodegeneration occurring not only in MD, but also in major neurodegenerative disorders such as AD or Parkinson’s disease. 

## 2. Results

### 2.1. Auditory Function Analysis

Profile of Mood States (POMS) analysis (Table 1) revealed in Group A subjects, the group treated with mushroom preparation, a significant improvement of subjective parameters related to the psycho-emotional status of the patients, as compared to untreated MD patients (Group B), where we did not observe particular changes. Table 2 shows homogeneity between the two groups regarding the number of crises, their duration, and the frequency of symptoms. Notably, data in Table 3 illustrates the Tinnitus Handicap Inventory (THI) questionnaire, performed to define the clinical grading of tinnitus severity, showing a statistically significant improvement in the group of patients receiving *Coriolus* mushroom biomass treatment, as compared to the untreated group.

To document SNHL, we performed in all subjects, at the initial (T0) phase, tonal audiometry analysis (Figure 1). For both experimental groups, the tonal interest was centered on medium-high frequencies, with an average intensity of 55 dB loss. All subjects in the group A reported in the T1 phase, after treatment, significant changes, both in the frequency range, and in the average loss in dB, as compared to the initial T0 phase. Similarly, speech audiometry analysis revealed in the same subjects receiving mushrooms a significant improvement of intellection threshold, i.e., the ability of verbal discrimination, with respect to the initial T0 phase, where the threshold of intellection and perception that is 100% of the given words was assumed to be 75 db. In contrast to the *Coriolus* biomass-treated group, in patients of Group B, however, we did not detect any significant change compared to thresholds measured at T0 initial phase. This finding was consistent with impedenzometric measures at examination, which revealed in all subjects either at T0 initial phase or at the T1 phase, an average increase in the threshold of stapedial reflexes and the positivity of the Metz test, indicative of cochlear suffering, with no significant differences between the two groups.

### 2.2. Redoxomics

#### Modulation of Hsp72, HO-1, Thioredoxin, Sirtuins and γ-GC Liase, in MD Patients after Coriolus Mushroom Supplementation

Oxidative stress plays a role in the pathogenesis of a wide variety of pathological states [35,36,37]. ROS can oxidize membrane lipids generating lipid hydroperoxides and many aldehydes such as HNE, luminescent by-products, and isoprostanes. HNE can accumulate in cells in relatively high concentrations and cause cell toxicity. Recent studies have also shown that in response to environmental changes and other stressful conditions promoting proteotoxicity [38,39,40], cells adaptively activate synthesis and accumulation of several members of stress proteins, primarily Hsp70 and HO-1. As reported in Figure 2 and Figure 3, mushroom supplementation with *Coriolus* biomass preparation resulted in up-regulation of the inducible isoforms of both Hsp70 and heme oxygenase-1 (HO-1), in lymphocytes (Figure 2a and Figure 3a), a finding observed also in plasma, (Figure 2b and Figure 3b), as compared to untreated group of MD patients. A representative Western blot obtained probing tissue samples with an antibody specific for the inducible isoform of heat shock proteins 70 (Hsp72) or Heme oxygenase are shown in Figure 2c,d and Figure 3c,d, respectively. Western blot analysis of the Thioredoxin protein also revealed a significant increase in the group of patients treated with *Coriolus* compared to control group, in lymphocyte and plasma (Figure 4a,b). A representative blot of thioredoxin protein is reported in Figure 4c,d. Similar results were also obtained analyzing sirtuin-1 expression. As shown in Figure 5a, Sirtuin-1 immunoreactivity was higher in lymphocytes of a group of MD patients treated for 2 months with *Coriolus* than in the untreated MD group. Consistent with this, plasma sirtuin-1 levels were higher in MD patients supplemented with mushrooms, as compared to the MD group of patients alone (Figure 5b). Representative blots of sirtuin-1 protein are reported in Figure 5c,d, respectively. Another important redoxomic component of vitagene network is γ-GC liase, the rate-limiting enzyme for intracellular glutathione (GSH) synthesis. Notably, GSH concentration and γ-GC liase activity are declining with age in the central nervous system (CNS), a condition associated with increased oxidative stress [41]. Here we report that lymphocyte γ-GC liase levels were higher in MD patients supplemented with mushrooms as compared to the MD group of patients alone (Figure 6a). A representative blot of γ-GC protein is reported in Figure 6b. 

### 2.3. Assessment of Systemic Oxidative Status

Protein and lipid oxidation occurring because of oxidative stress in tissues and organs leads to the formation of carbonyl groups in amino acid residues [41] and, respectively, to 4-hydroxynonenal (HNE) formation from arachidonic acid or other unsaturated fatty acids [42]. As a hallmark for oxidative damage to proteins by free-radical attack, protein carbonylation, by binding via Michael addition to proteins, particularly at cysteine, hystidine, or lysine residues [36], exerts deleterious effects on cell function and viability, being generally unrepairable and leading to production of potentially harmful protein aggregates and to cellular dysfunction. Under conditions of oxidative stress, protein oxidation products measured as protein carbonyls, as well as lipid oxidation products, measured by HNE or ultraweak luminescence, accumulate [6,29,30]. Examination of plasma protein carbonyls (Figure 7a) and HNE (Figure 7b), as well as plasma or lymphocyte ultraweak luminescence levels (Figure 7c) revealed a significant elevation in MD patients respect to *Coriolus*-treated group of MD patients.

### 2.4. Lipidomics Analysis

Oxidation of polyunsaturated fatty acid arachidonic, eicosapentaenoic, docosahexaenoic, linoleic, and dihomo-γ-linolenic generate bioactive lipids. The development of mass spectrometry platforms enabling quantification of diverse lipid species in human urine is of crucial importance to understand metabolic redox homeostasis in normal as well as pathophysiological conditions. Here we demonstrate clearly how administration of *Coriolus* to MD patients increases significantly the powerful anti-inflammatory eicosanoid LXA4 in plasma and lymphocytes as compared to untreated MD patients (Figure 8a,b). The same results were observed in urine, where a large increase in LXA4 was measured after *Coriolus* supplementation (Figure 8c). Consistently, analysis of urine levels of pro-inflammatory eicosanoids 11-dehydro TXB2, isoprostane PGF2α, and isoprostane iPF2α-VI showed the opposite results with significantly higher levels of these bioactive lipids in MD subjects than the levels found in *Coriolus* administered MD patients (Figure 9a–c).

Consistent with other findings showing that oxidative stress and altered thiol status in degenerating brain diseases correlates with systemic redox imbalance and oxidative stress, as in AD [31,32,33,34], the content of total GSH, reduced and oxidized glutathione and the GSH/GSSG ratio, was determined in the plasma of MD patients as a measure of the antioxidant status and compared with the levels of *Coriolus*-treated MD group (Table 4). We report the plasma from MD patients contained significantly lower levels of GSH as compared to *Coriolus*-supplemented patients, which paralleled to corresponding significantly higher GSSG levels (*p* < 0.05) (Table 4). These changes resulted in a plasma GSH/GSSG ratio which was significantly higher in the group of MD plus *Coriolus* subjects then the ratio found in the MD group alone (Table 4).

## 3. Discussion

MD is a chronic illness derived from combined neurodegenerative events occurring at level of spiral ganglion as well as hair cells of the inner ear, associated with a negative impact on the quality of life of individuals, presenting various symptoms, such as temporary hearing loss, dizziness, and tinnitus [7]. 

After its initial description by Prosper Meniere more than 150 years ago, the disease named after him is still at the center of scientific debate [1]. MD is a hydropic ear pathology, where episodic vertigo, tinnitus, and fluctuating hearing loss coexist with endolymphatic hydrops. [3]. Recent evidence indicates the involvement of oxidative stress in the development of endolymphatic hydrops associated with neuronal ganglion damage with apoptotic neuronal cell death as a prominent factor contributing to SNHL found in the later stages of MD [2]. Thus, it is conceivable that MD, as a systemic oxidant disorder [5], can be also considered, owing to its demonstrated neurodegenerative nature of the neuronal cochlear ganglion component, involved in its pathogenesis, a pursuable investigative model of neurodegeneration. Consistent with this possibility, the present study was undertaken to explore the hypothesis that changes in the redox status of glutathione, stress-responsive vitagenes, and lipidomics, the major determinants in the disruption of redox homeostasis affecting spiral ganglion neurons, may be positively impacted by nutritional intervention with *Coriolus*-MRL biomass supplementation.

Mushrooms have been present in traditional medicine for thousands of years, and are reportedly endowed with immunomodulatory actions, associated with antioxidant, anticancer, antiviral, bacteriostatic, and hepatoprotective properties [43]. Mushroom-derived therapeutics, mainly polysaccharopeptides isolated from *Coriolus versicolor*, are well characterized and commercially available. Here we tested the hypothesis that neurotoxicity is an important causative factor involved in MD pathogenesis, which can be evaluated by measuring markers of oxidative stress and cellular stress response proteins in the peripheral blood of patients with MD. We evaluated in the present study systemic oxidative stress and cellular stress response in 40 patients suffering from MD in the absence and in the presence of treatment with mushroom biomass preparation from *Coriolus*. Systemic oxidative stress was estimated in plasma and urines of patients with MD or MD plus *Coriolus*, by measuring protein carbonyls, HNE, and ultraweak luminescence, as well as active biolipids such as lipoxin A4 and F2- isoprostanes, whereas in the lymphocyte heat shock proteins (HSP) heme oxygenase-1 (HO-1), Hsp70 and thioredoxin (Trx) levels were measured to evaluate the systemic cellular stress response. Increased levels of DNPH, HNE, ultraweak luminescence, and F2-isoprostanes were found in all the samples from MD patients with respect to the MD plus *Coriolus*-treated group. This was paralleled by a significant induction of lymphocyte HO-1, Hsp70, TrxR-1 as well as Sirtuin-1 and by a significant increase in the plasma ratio-reduced glutathione GSH) vs. oxidized glutathione (GSSG).

It is suggested that genetic factors may contribute partly to the etiologies of MD, as some associations have been reported for polymorphisms related to gene coding for protein involved in inflammation, circulation, and blood vessels, such as interleukin 1A (–889C/T), interleukin 6–572C/G), protein kinase C beta type (1425G/A), matrix metalloproteinase-1 (–1607G/2G), methylenetetrahydrofolate reductase (MTHFR) (C677T), prothrombin (G20210A), and complement factor H [44,45,46,47,48,49,50,51], and genes involved in free-radical processes. Although the initial causative factors triggering the disease have not been clarified, various genes and variants have been confirmed to be related to MD, which also suggests a specific family of genetic predisposition and implies genetic factors as key players in the initiation and progression of MD [52]. Consistent with this scenario, inflammation and oxidative stress-induced endolymphatic hydrops have been identified as a secondary pathogenesis of the disease [53]. Thus, MD etiology and pathogenesis appears to be an aberrant response of the adaptive or innate immune system, ultimately mediated by pro-inflammatory and oxidative processes underlying its physiopathological determinism [54]. Several mechanisms are involved in the development of immune-mediated inner-ear pathology, including (a) similarity with potentially harmful component of virus or bacteria, such as cross-reactive epitope inducing inner-ear damage; and (b) generation of pro-inflammatory Interleukin 1 β (IL-1B) or Tumor necrosis factor α (TNF) cytokines and transcriptional nuclear factor kB (NF-kB) [55]. Toll-like receptor coding genes, including TLR3, TLR7, TLR8, and TLR10, are widely reported to contribute to the disease, being directly related to the initiation and progression of MD, thus implying a specific role for the immune system during the pathological processes [56]. This is confirmed by recent findings highlighting the relationships between increased serum levels of IL6 and IL1 with vertigo, a specific complication of MD. 

To survive different types of injuries and adapt to environmental changes, neuronal cells have evolved networks of responses capable of detecting and controlling different forms of stress [26,27,28,29,30,31,32,33,34,35,36,37,38,39,40,41,42,43,44,45,46,47,48,49,50,51,52,53,54,55,56,57,58,59,60,61,62]. As such, integrated survival mechanisms exist in the brain based on the activity of redox-dependent genes, called vitagenes, capable of sensing stress and including HSP (Hsps), thioredoxin, γ-GC ligase, and sirtuin family proteins, which together with bioactive lipids represent the last step in the “omic” cascade starting from genome, through transcriptome, proteome, and finally to metabolome. Lipid mediators as signaling factors play a fundamental role in the initiation, amplification, and resolution of inflammation [33,34]. Thus, use of urine sample for lipidomic analysis enables reproducible quantification of several lipid metabolites generated by lipoxygenase, cyclooxygenase, and cytochrome P450 activities, such as octadecanoids, eicosanoids, and docosanoids. Lipidomic analysis of urine reveals quantitative data that reflects the alterations in in eicosanoids levels seen in MD patients as compared to normal controls. Lipoxin A4, in particular, is a metabolic product of arachidonic acid, acting as an endogenous ‘‘breaking signal’’ towards inflammatory processes, actively operating in the detection and control of diverse forms of stress in the brain. Owing to its potent anti-inflammatory properties LXA4 positively influences the outcome in many inflammatory disorders, such as nephritis, periodontitis, arthritis, and inflammatory bowel disease [63,64]. Chronic inflammation sustains the progression of neurodegenerative pathologies, including AD and Parkinson’s disease, but also in specific neuronal districts, as in the cochleovestibular apparatus and the spiral ganglion neural cells. Thus, identification of mechanisms capable of favorably impacting the pro-inflammatory environment generated in the MD pathology represents an area of active investigation. Consistent with this notion, the activation of the LXA4 pathway could therefore serve as a potential therapeutic target to treat MD-associated inflammation and cochleovestibular dysfunction. As LXA4 action is mediated by LXA4 receptor (ALX), a formyl-peptide receptor-like 1 (FPRL1) present on cellular membrane [33], the discovery of agents with the potential of increasing Lipoxin A4 (LXA4), and consequently of reducing inflammatory-mediated endolymphatic hydrops, can be relevant to therapeutics of this disease. Eicosanoid lipoxin A4 (LXA4) decreases toxic compounds such as ROS, inhibits recruitment of activated neutrophils and blocks accumulation of pro-inflammatory cytokines, thereby promoting resolution of inflammation [65]. 

Our results obtained with a nutritional approach based on a *Coriolus versicolor* biomass supplementation are relevant to innovative therapeutic anti-inflammatory strategies aimed to minimize consequences associated with neurodegeneration and oxidative stress of cochleovestibular system pathologies including not only MD but also sudden sensorineural hear loss (SSNHL) where it has recently demonstrated a critical role played by NLRP3 inflammasome [41]. NLRP3 is a sensor of the intracellular innate immune response expressed in immune cells, including monocytes and macrophages. Activation of the NLRP3 inflammasome results in augmentation of IL-1β secretion and cochlear autoinflammation. [66,67].

Due to different biological routes of actions, ranging from anticancer, antiviral, bacteriostatic, and regulation of of immune function, as well as antioxidant and protectant of hepatocytes [29,68], relevant to the inflammatory disease pathogenesis, mushrooms in the past have been diffusely applied for therapeutic use in traditional medicine [27,28]. It has been shown, in fact, that cytokine response triggered by activated immune cells occurs after stimulation with immunostimulatory molecules derived from mushroom preparations, which are mainly β-glucans [30,57,69,70,71]. Despite this, however, the active ingredients are not fully characterized, which makes mushroom extracts very difficult to reconcile with current pharmaceutical practices involving highly purified compounds and, therefore, difficult to patent, as they are complex mixtures of molecules of unknown concentrations to be administered for therapeutic purposes. In addition, mushroom-derived polysaccharides are complex molecules that cannot be synthesized, as the mass production of these compounds would require timely and costly extraction processes. Consequently, most research efforts have focused on low molecular weight compounds, such as cordycepin [72], which is a cytotoxic nucleoside analog inhibitor of cell proliferation. However, polysaccharopeptides isolated from *Coriolus versicolor* are well characterized and their commercial diffusion well established. In addition to its medical applications, *Coriolus versicolor* is widely used to degrade organic pollutants such as pentachlorophenol (PCP) [73]. Notably, as previously mentioned, several studies have demonstrated significant ultrastructural reductions in dendritic innervation densities, at level of cochlear ganglion neurons, pointing the possibility that neurotoxicity plays an important role in the pathology of MD [6]. Interestingly, a recent study in mice has found that *Coriolus versicolor* biomass promotes significant increases in dendritic length and branching and total dendritic volume of immature neurons, suggesting a positive effect of oral *Coriolus versicolor* administration on hippocampal neurogenic reserve [74]. Taking all this into account and given the inflammatory pathogenesis of MD degenerative damage, our findings of reduction in oxidative stress and inflammatory mediators associated with increased anti-inflammatory metabolites in mushroom-treated patients has innovative therapeutic potential. 

Moreover, increasing evidence suggests that alteration of redox status, overloading of peroxidative product hydroxynonenals (HNE) or protein carbonyls can severely alter redox homeostasis [1]. Thus, the ensuing oxidative stress is a primary causative factor underlying endolymphatic hydrops pathogenesis, associated with cellular degenerative damage and apoptotic cell death affecting vulnerable cells of cochleovestibular apparatus, and thus contributing to the SNHL and vestibular dysfunction found in later stages of MD. 

Moreover, it is known that normal auditory function depends on maintenance of the unique ion composition in the endolymph. Hence, reduction of microvascular alterations due to decreased oxidative stress in the inner ear after mushroom treatment has relevant implications [34]. Our data on the modulation of the stress-responsive protein involved in stress tolerance and cell survival are relevant as a potential target of mushrooms therapeutics and nutritional redox approaches, as the ability of neurons to cope with stressful conditions relies upon the capability to activate stress-responsive pro-survival pathways that normally function at a very low level and that result generally in increased synthesis of antioxidant and anti-apoptotic molecules. Among the cellular pathways conferring protection against oxidative stress, a key role is played by vitagenes, which include HSP (Hsps) Hsp70, heme oxygenase-1, and small Hsps, together with thioredoxin, enzymes of Meister cycle for the synthesis of glutathione and sirtuins [10,24]. Given the broad cytoprotective properties of the heat shock response there is now emerging interest in developing pharmacological agents able to potentiate neuroprotective stress responses [26]. When appropriately activated, cellular stress response can restore redox equilibrium and neuronal homeostasis.

## 4. Materials and Methods

### 4.1. Chemicals

5,5’-Dithiobis-(2-nitrobenzoic acid) (DTNB), 1,1,3,3-tetraethoxypropane, purified bovine blood SOD, NADH, glutathione (GSH), glutathione disulfide (GSSG), nicotinamide adenine dinucleotide phosphate (β-NADPH, type 1, tetrasodium salt), and glutathione reductase (GR; Type II from Baker’s Yeast), were from Sigma Chemicals Co, St. Louis (USA). All other chemicals were from Merck (Darmstadt, Germany) and of the highest grade available. 

### 4.2. Coriolus Versicolor Biomass Preparation

*Coriolus versicolor* is found almost worldwide; however, its bioactivity varies depending on the habitat in which it grows. To eliminate these variations, established CV-OH1 strain was used which demonstrates rapid and aggressive colonization. According to the manufacturer procedure (Mycology Research Laboratories Ltd., Luton, UK) *Coriolus versicolor* containing both mycelium and primordia (young fruit body) biomass, obtained cultivating the biomass that is grown on a sterilized (autoclaved) substrate. The production process involves the inoculation of sterile organic edible grain with spawn from the mother culture. The fungus is allowed to completely colonize the growth medium aseptically. At the correct stage of development, corresponding to the maximum bioavailability the living biomass is aseptically air-dried, granulated, tested microbiologically, and reduced in powder for tablet preparation. In comparison to *Coriolus* extracts, biomass has the advantage of preserving all nutraceutical potential which is usually reduced with extracts or concentrates, including lyophilization, and thus the activity of the product corresponds with the source mushroom, while being further intensified by using the entire mycelium. Tablets of 500 mg each of the *Coriolus* biomass containing mycelium and primordia of the respective mushroom, kindly provided by Mycology Research Laboratories Ltd. (MRL, Luton, UK), as the product commercially available, were used for experiments. Optimal dosage (200 mg/kg) was chosen according to the dose used in clinical trials with cancer or Human papilloma virus (HPV) patients (3 g/day) [57], a regimen also confirmed by studies in rat [33].

### 4.3. Ethical Permission

The study was approved by the local Ethics Committee (prot. N. 76/2018/PO, 16 April 2018) and informed consent was obtained from all patients. 

### 4.4. Patients

We enrolled 40 patients (22 males and 18 females, with an average age of 49.5 +/− 14.6 years; range 29–60 years) with MD according to the diagnostic scale of the Committee on Hearing and Equilibrium of the American Academy of Otolaryngology—Head and Neck Surgery published in 1995 for MD [13,58] (two or more definitive spontaneous episodes of vertigo 20 min or longer, audiometrically documented hearing loss on at least one occasion, tinnitus or aural fullness in the treated ear). Patients were divided into two groups, A and B. Group A consisted of 22 patients suffering from cochlear sensorineural hearing loss (SNHL) that was been subjected to treatment with biomass preparation from *Coriolus versicolor* mushroom (MRLs), administered orally in tablets of 500 mg (3 tablets every 12 h, morning and evening, for 2 consecutive months), while Group B, formed of 18 patients, also suffering from cochlear SNHL, was not subjected to any treatment. Constituted exclusion criteria: (i) older than 60 years; (ii) presence of cardiovascular diseases; (iii) presence of metabolic disorders and/or parts; (iv) the presence of external ear pathologies and/or medium; (v) presence of alterations of state-acoustic nerve; (vi) prior learning and/or recent treatment with antioxidant drugs or otherwise active in the compartment cochlear.

All patients, after targeted anamnestic investigation, underwent the T0 initial phase, where the Profile of Mood States (POMS) questionnaire was administered, to assess the emotional and degree of psychological stress status, indexed on the basis of specific elements, such as: Tension–Anxiety (TA), Depression–Discouragement (D), Anger–Hostility (AH), Vigor–Activity (V), Fatigue (F), Confusion–Loss (C), in relation to the impairment caused in each subject from hearing impairment. The POMS original scale contains 65 self-report items using the 5-point Likert Scale. Participants can choose from 0 (not at all) to 4 (extremely). In addition, all subjects were given a tinnitus questionnaire consisting of 40 multiple choice questions to define the impact of symptoms on the patient life. In the groups the grade of severity for each patient was established on the basis of the vertigo attack frequency over a year (from 2 to 8 crisis), the intensity and the duration of symptoms (from a few days, to some weeks, to a month in the most severe case). In addition, the hearing loss degree was assessed instrumentally, allowing staging of the disease in MD patients. 

Enrolled patients were also examined to define the qualitative and quantitative characteristics of auditory function: (a) examination ENT; (b) test tone audiometry; (c) speech audiometry; (d) impedenzometry examination. Such instrumental examinations were aimed at defining not only the extent of hearing, but also the location of the SNHL, to define it whether cochlear or retrocochlear. Each patient, either in Group A or Group B, was subjected to blood and urine sampling for biochemical analysis in plasma and lymphocytes, and in urines of specific markers of cellular oxidative stress, lipid and protein oxidative metabolism, cellular stress response (vitagenes), glutathione status (reduced glutathione (GSH), oxidized glutathione (GSSG), and GSH/GSSG ratio and lipoxin A4. Phase T1 in Group A, the mushroom-treated group of patients, was accomplished by oral administration of MRL *Coriolus versicolor* biomass compound for 2 consecutive months, to assess its neuroprotective, anti-neuroinflammatory potential, and thus test the possible protection against the cellular degeneration in general and, particularly, in the inner ear. At 2 months from the beginning of treatment (T1 phase) we evaluated in all patients the degree of evolutive trend of auditory function as well as cellular oxidative stresses, redox status, cellular stress response, and Lipoxin A4, in the blood. The correlative analysis, aimed at highlighting the antioxidant properties of the compound administered and its effects at the cellular level as well as at the clinical level, audiological function, will define a neurobiological clinical model to assess the effectiveness of pharmacological compounds in counteracting oxidative stress and neuroinflammatory damage associated with MD.

### 4.5. Sampling

Blood (6 mL) was collected after an overnight fast by venopuncture from an antecubital vein into tubes containing ethylenediamine tetraacetic acid (EDTA) as anticoagulant. Immediately after sampling, two blood aliquots were separated: first 2 mL were centrifuged at 10.000× *g* for 1 min at 4 °C to separate plasma from red blood cells; the remaining aliquot (4 mL) was used for lymphocytes purification. All samples were stored at −80 °C until analysis.

### 4.6. Lymphocytes Purification

Lymphocytes from peripheral blood were purified by using the Ficoll Paque System following the procedure as suggested by the manufacturer (GE Healthcare, Piscataway, NJ, USA).

### 4.7. Western Blot Analysis

Plasma samples were processed as such, while the isolated lymphocyte pellet was homogenized and centrifuged at 10,000× *g* for 10 min. The supernatant was then used for analysis after determination of protein content. Proteins extracted for each sample, at equal concentration (50 μg), were boiled for 3 min in sample buffer (containing 40 mM Tris-HCl pH 7.4, 2.5% SDS, 5% 2-mercaptoethanol, 5% glycerol, 0.025 mg/mL of bromophenol blue) and then separated by SDS-polyacrylamide gel electrophoresis (SDS-PAGE). Separated proteins were transferred onto nitrocellulose membrane (BIO-RAD Hercules, CA, USA) in transfer buffer containing 0.05% of SDS, 25 mM di Tris, 192 mM glycine and 20% v/v methanol. The transfer of the proteins on the nitrocellulose membrane was confirmed by staining with Ponceau Red which was then removed by 3 washes in PBS (phosphate buffered saline) for 5 min/each. Membranes were then incubated for 1 h at room temperature in 20 mM Tris pH 7.4, 150 mM NaCl and Tween 20 (TBS-T) containing 2% milk powder and incubated with appropriate primary antibodies, namely anti-γ-GC liase anti-Hsp70, anti-HO-1, anti-Sirt-1, anti-Trx and anti-HNE polyclonal antibody (Santa Cruz Biotech. Inc.), overnight at 4 °C in TBS-T. The same membrane was incubated with a goat polyclonal antibody anti-beta-actin (SC 1615 Santa Cruz Biotech. Inc., CA, USA, dilution 1:1000) to verify that the concentration of protein loaded in the gel was the same in each sample. The excess of unbound antibodies was removed by 3 washes with TBS-T for 5 min. After incubation with primary antibody, the membranes were washed 3 times for 5 min in TBS-T and then incubated for 1 h at room temperature with the secondary polyclonal antibody conjugated with horseradish peroxidase (dilution 1:500). The membranes were then washed 3 times with TBS-T for 5 min. Finally, the membranes were incubated for 3 min with SuperSignal chemiluminescence detection system kit (Cod 34080 Pierce Chemical Co, Rockford, USA) to display the specific protein bands for each antibody. The immunoreactive bands were quantified by capturing the luminescence signal emitted from the membranes with the Gel Logic 2200 PRO (Bioscience) and analyzed with Molecular Imaging software for the complete analysis of regions of interest for measuring expression ratios. The molecular weight of proteins analyzed was determined using a standard curve prepared with protein molecular weight.

### 4.8. Glutathione and Glutathione Disulfide Assay

GSH and GSSG were measured by the NADPH-dependent GSSG reductase method as previously reported in Calabrese et al. 2010. Lymphocytes were homogenized on ice for 10 s in 100 mM potassium phosphate, pH 7.5, which contained 12 mM disodium EDTA. For total glutathione, an aliquots (0.1 mL) of homogenates were immediately added to 0.1 mL of a cold solution containing 10 mM DTNB and 5 mM EDTA in 100 mM potassium phosphate, pH 7.5. The samples were then mixed by tilting and centrifuged at 12,000× *g* for 2 min at 4 °C. An aliquot (50 µL) of the supernatant was added to a cuvette containing 0.5 U of GSSG reductase in 100 mM potassium phosphate and 5 mM EDTA, pH 7.5 (buffer 1). After 1 min of equilibration, the reaction was initiated with 220 nmol of NADPH in buffer 1 for a final reaction volume of 1 mL. The formation of a GSH-DTNB conjugate was then measured at 412 nm. The reference cuvette contained equal concentrations of DTNB, NADPH, and enzyme, but not sample. For assay of GSSG, aliquots (0.5 mL) of homogenate were immediately added to 0.5 mL of a solution containing 10 mM N-ethylmaleimide (NEM) and 5 mM EDTA in 100 mM potassium phosphate, pH 7.5. The sample was mixed by tilting and centrifuged at 12,000× *g* for 2 min at 4 °C. An aliquot (500 µL) of the supernatant was passed at one drop/s through a SEP-PAK C18 Column (Waters, Framingham, MA) that had been washed with methanol followed by water. The column was then washed with 1 mL of buffer 1. Aliquots (865 µL) of the combined eluates were added to a cuvette with 250 nmol of DTNB and 0.5 U of GSSG reductase. The assay then proceeded as in the measurement of total GSH. GSH and GSSG standards in the ranges between 0 to 10 nmol and 0.010 to 10 nmol, respectively, added to control samples were used to obtain the relative standard curves, and the results were expressed in nmol of GSH or GSSG, respectively, per mL or mg protein.

### 4.9. Spontaneous Ultraweak Chemiluminescence Assay

Measurement of chemiluminescence in blood samples was accomplished according to the method of Flecha et al. 1991 [59]. Briefly, aliquots (0.5 mL) of plasma were diluted 1:1 with 30 mM phosphate buffer (pH 7.4), whereas lymphocyte pellet was homogenized and centrifuged at 10,000× *g* for 10 min. Before aliquots (0.5 mL) of the supernatant were taken and diluted 1:1 with 30 mM phosphate buffer (pH 7.4) at 0–4 °C and centrifuged at 10,000 *g* for 3 min at 0–4 °C. Then spontaneous ultraweak chemiluminescence (UCL) was measured in the supernatant at 30 °C with a Turner TD 20/20 luminometer. The sensitivity was adjusted to 50%, and results were expressed as luminescence units/mg protein.

### 4.10. Lipidomic Analysis

For oxylipins determination plasma and urine samples were extracted essentially as described by Wolfer et al. 2015 [60]. Briefly plasma samples were prepared by transferring 100 μL to the preparation plate after thawing and brief vortexing. A volume of 20 μL of IS working solution and 30 μL of 2% formic acid solution in water were added, and the plate was capped and gently mixed. The SPE plate was conditioned using 200 μL of MeOH and the sorbent equilibrated with 200 μL of H_2_O. Samples were transferred from the preparation plate to the SPE, the preparation plate was further washed with 50 μL of MeOH/H_2_O 1:1, and the rinsing solution will be added to the SPE plate. Following aspiration, the SPE plate will washed with 200 μL of H_2_O + 2% NH_4_OH and 200 μL of H_2_O/ACN 1:1. Oxylipins will be eluted with 4 × 25 μL of MeOH + 2% formic acid. The elution fraction was evaporated under N_2_ and the residues reconstituted in 120 μL of MeOH/H_2_O. Urine samples were prepared by mixing 50 μL of urine with 50 μL of MeOH and 20 μL of IS working solution following thawing and brief vortexing.

Oxylipins determination was carried out with ultrahigh performance liquid chromatography (UHPLC) coupled with mass spectrometry (triple quadrupole, q-tof or orbitrap instrument) the methods and conditions of ionization were chosen to achieve the best results in order of quantification and reproducibility.

### 4.11. Lipoxin A4 Assay

LXA quantification was performed using an enzyme-linked immunosorbent assay (ELISA) kit following the protocol provided by the company. Biological fluid rates were used, and the measurement performed by a spectrophotometer at a wavelength of 450 nm.

### 4.12. Determination of Protein

Proteins were estimated by the bicinchoninic acid (BCA) protein assay method [61], using bicinchoninic acid reagent.

### 4.13. Statistical Analysis

Results were expressed as means ± SEM of *n* = 18 experiments (MD alone) or *n* = 22 experiments (MD plus *Coriolus*), each of which were performed, unless otherwise specified, in triplicate. Data were analyzed by one-way Analysis of Variance (ANOVA), followed by inspection of all differences by Duncan’s new multiple-range test. Differences were considered significant at *p* < 0.05. 

## 5. Conclusions

Brain cells, such as spiral ganglion neurons, possessing relatively weak endogenous antioxidant potential, show a particular need for activation of antioxidant pathways, which becomes a central prerequisite under conditions of oxidant insults, such those underlying not only the pathogenesis of MD but also acting in a broad range of age-associated diseases. Aging, in fact, is based on complex mechanisms and systemic processes, whose major gap remains insufficient knowledge about the proactive pathway shift from normal “healthy” aging to disease-associated pathological aging [75]. As a major complication of normal “healthy” aging, the increased risk of age-related diseases, such as cancer, diabetes mellitus, cardiovascular and neurodegenerative diseases, including cochleovestibular dysfunctions that can adversely affect the quality of life in general, with enhanced incidence of co-morbidities and mortality, should be considered.

We evaluated systemic oxidative stress and cellular stress response in MD patients in the absence and in the presence of treatment with a biomass preparation from *Coriolus versicolor*. It was concluded that systemic oxidative stress was reduced in MD patients treated with *Coriolus versicolor*, which was paralleled by a significant induction of vitagenes and by an increased plasma GSH vs. GSSG ratio. Vitagene up-regulation after *Coriolus versicolor* supplementation indicates a maintained response to counteract intracellular pro-oxidant status. In a contextualized global “omics” approach, the combination of redoxomics and lipidomics, being more stable than the metabolome and closer to the phenotype than the transcriptome, represents the most promising “omics” field enabling dissection and perhaps full comprehension at molecular and cellular level, of aging mechanisms and age-related processes.

Approaching the redox biology of the aging inner-ear system, as exploited in the present study, together with broadening of the potential of lipidomic analysis represents an innovative tool for monitoring at the omic level to the extent of oxidative insult and related modifications, allowing the identification of targeted antioxidative cytoprotective vitagene system proteins. The present study also highlights the importance of investigating MD as a convenient model of cochlear neurodegenerative disease.

## Figures and Tables

**Figure 1 ijms-21-00284-f001:**
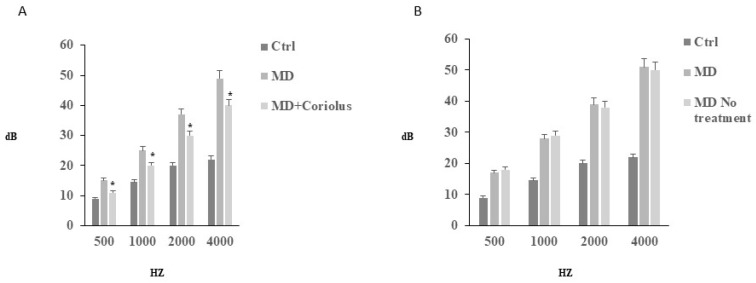
Tonal audiometry analysis. Tonal interest was centered on medium–high frequencies, with an average intensity of 55 dB loss. All subjects reported in both T0 (**B**) and T1 (**A**) phases no significant changes, either in the frequency range, or in the average loss in dB. Speech audiometry analysis, revealed in subjects of group A, who received mushrooms, a significant improvement of intellection threshold, i.e., the ability of verbal discrimination, respect to the initial T0 phase, where the threshold of intellection and perception that is 100% of the given words, was assumed to be 75 db.

**Figure 2 ijms-21-00284-f002:**
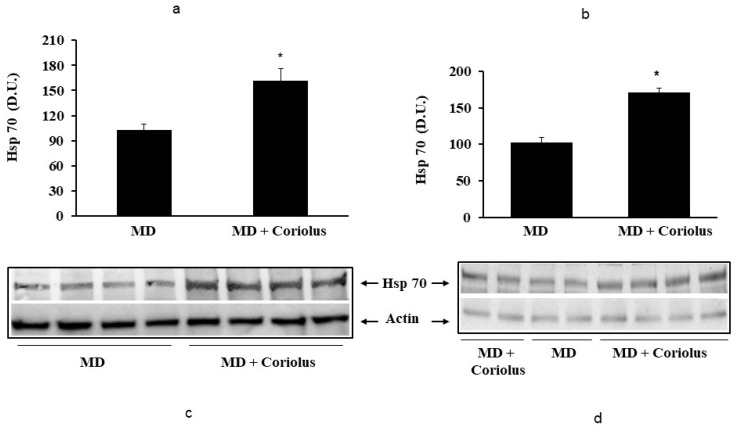
Heat shock protein 70 levels in lymphocytes and in plasma from MD patients. Samples from MD patients were assayed for heat shock protein 70 (Hsp70) by western blot as described in Materials and Methods. A representative immunoblot is shown in (**c**,**d**). β-actin has been used as loading control. The bar graphs (**a**,**b**) show the densitometric evaluation and values are expressed as mean ± SEM of independent analyses on 22 patients (MD plus *Coriolus* biomass) and, respectively, on 18 patients (MD alone), per group. * *p* < 0.05 vs. MD alone. D.U., densitometric units.

**Figure 3 ijms-21-00284-f003:**
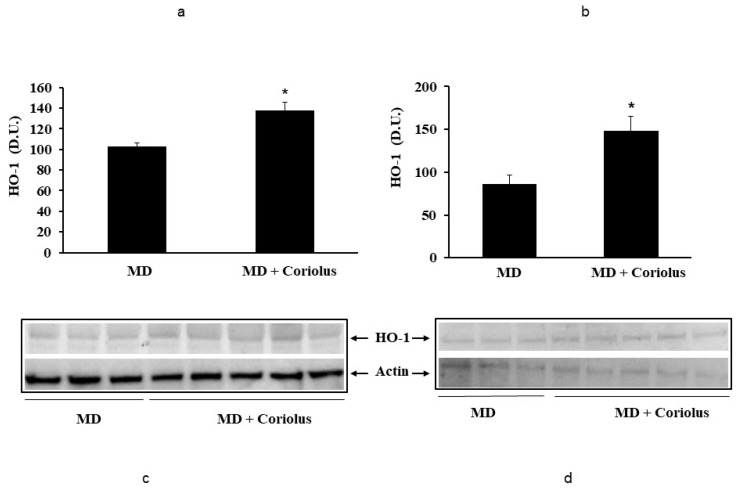
Heme oxygenase-1 levels in lymphocytes and in plasma from MD patients. Samples from MD patients were assayed for heme oxygenase-1 (HO-1) by western blot as described in Materials and Methods. A representative immunoblot is shown. β-actin has been used as loading control (**c**,**d**). The bar graph shows the densitometric evaluation (**a**,**b**) and values are expressed as mean ± SEM of independent analyses on 22 patients (MD plus *Coriolus* biomass) and, respectively, on 18 patients (MD alone), per group. * *p* < 0.05 vs. MD alone. D.U., densitometric units.

**Figure 4 ijms-21-00284-f004:**
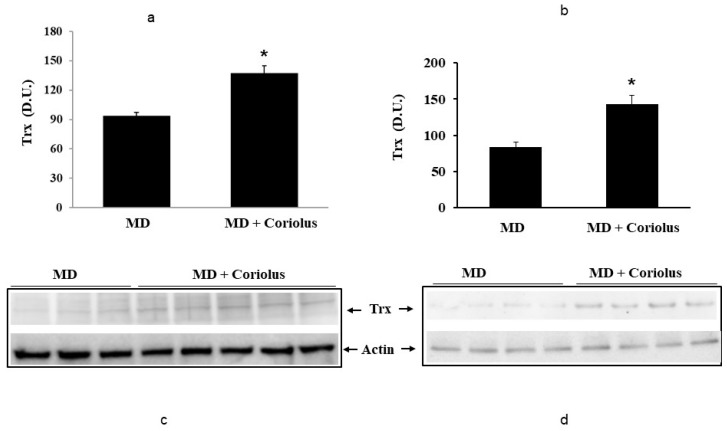
Thioredoxin levels in lymphocytes and in plasma from MD patients. Lymphocyte samples (**a**) and plasma samples (**b**) from MD patients were assayed for thioredoxin (Trx) by western blot as described in Materials and Methods. A representative immunoblot is shown (**c**,**d**). β-actin has been used as loading control. * *p* < 0.05 vs. MD alone. D.U., densitometric units.

**Figure 5 ijms-21-00284-f005:**
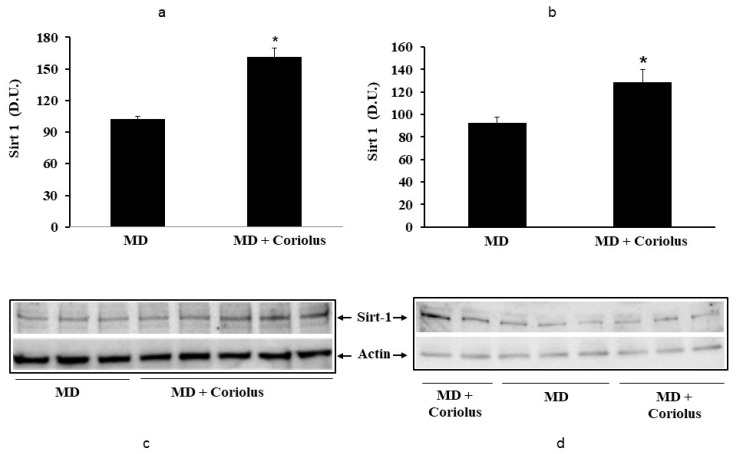
Levels of sirtuin-1 in lymphocytes (**a**) and plasma (**b**) from MD patients. Samples from MD patients were assayed for sirtuin-1 by Western blot as described in Materials and Methods. Representative immunoblots are shown in the same figure (**c**,**d**). β-actin has been used as loading control. The bar graph shows the densitometric evaluation and values are expressed as mean ± SEM of independent analyses on 22 patients (MD plus *Coriolus* biomass) and, respectively, on 18 patients (MD alone), per group. * *p* < 0.05 vs. MD alone. D.U., densitometric units.

**Figure 6 ijms-21-00284-f006:**
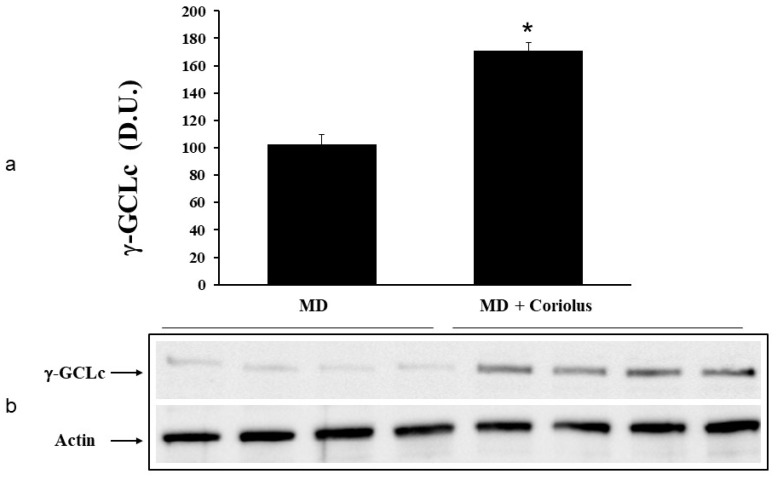
γ-GC liase levels in lymphocytes from MD patients. Plasma samples from MD patients were assayed for γ-GC liase by western blot as described in Materials and Methods. A representative immunoblot is shown (**b**). β-actin has been used as loading control. The bar graph shows the densitometric evaluation and values are expressed as mean ± SEM of independent analyses on 22 patients (MD plus *Coriolus* biomass) and, respectively, on 18 patients (MD alone), per group (**a**). * *p* < 0.05 vs. MD alone. D.U., densitometric units.

**Figure 7 ijms-21-00284-f007:**
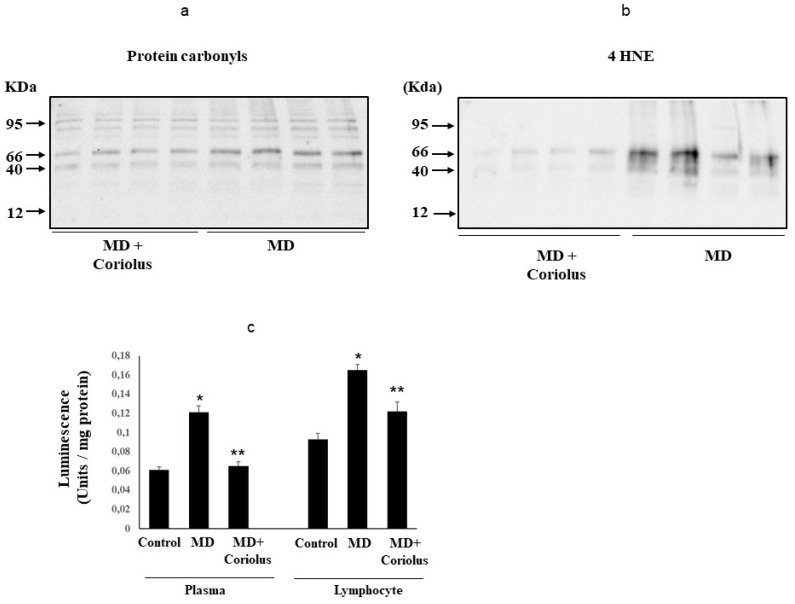
Protein carbonyls, 4-hydroxy-2-nonenals and Spontaneous ultraweak chemiluminescence (UCL) levels in MD patients. Plasma samples from MD patients (**a**,**b**) were assayed for protein carbonyls (DNPH) and 4-hydroxy-2-nonenals (HNE) by Western blot as described in Materials and Methods. Values are expressed as mean ± SEM of independent analyses on 22 patients (MD plus *Coriolus* biomass) and, respectively, on 18 patients (MD alone), per group. * *p* < 0.05 vs. MD alone. D.U., densitometric units. UCL in plasma and lymphocytes of control healthy volunteers and Meniere Diseased (MD) patients, in the absence and presence of Coriolus biomass treatment is shown in (**c**). UCL was measured as described in methods. CTRL: control; MD: Meniere disease patients. (*) *p* < 0.05 vs. control; (**) *p* < 0.05 vs. MD alone.

**Figure 8 ijms-21-00284-f008:**
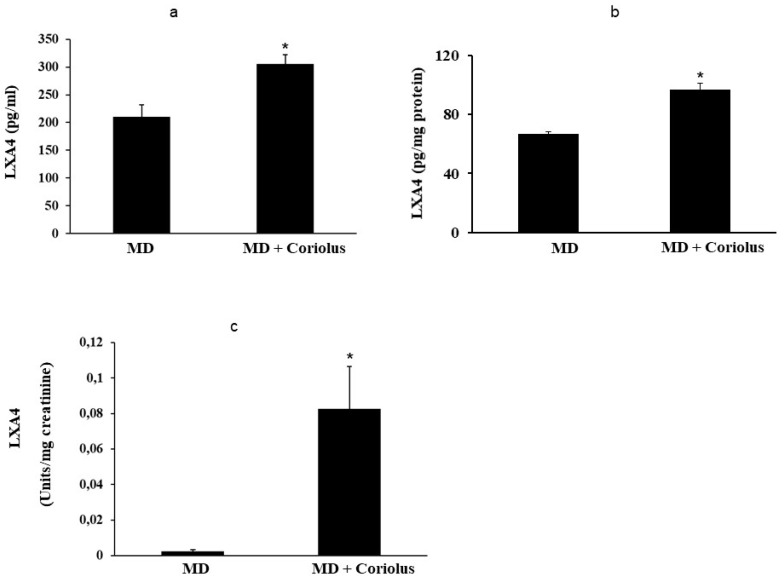
Lipidomic analysis of bioactive lipids. Biolipids are synthesized by oxidation of polyunsaturated fatty acids, arachidonic acid, eicosapentaenoic acid, docosahexaenoic acid, linoleic acid, and dihomo-γ-linolenic acid. The development of enabling mass spectrometry platforms for the quantification of diverse lipid species in human urine is of paramount importance for understanding metabolic redox homeostasis in normal and pathophysiological conditions. Anti-inflammatory eicosanoid LXA4 were measured in plasma, lymphocytes (**a**,**b**) and in urine (**c**), as compared to untreated MD patients.

**Figure 9 ijms-21-00284-f009:**
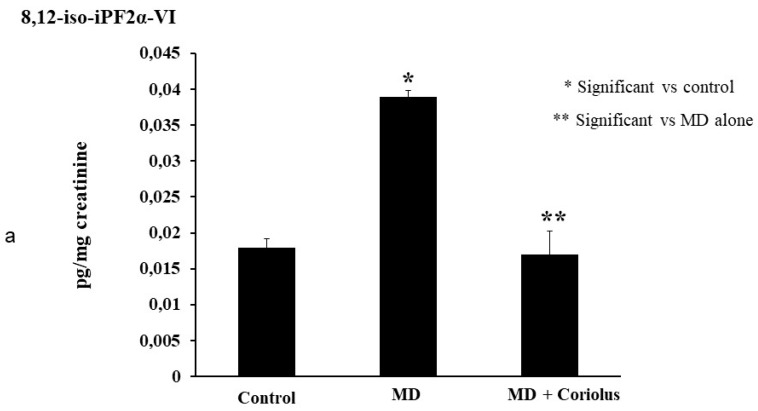
Lipidomic analysis of bioactive lipids. Analysis of urinay pro-inflammatory eicosanoids, 11-dehydro TXB2, isoprostane PGF2α, isoprostane iPF2α-VI, showing opposite results with significant higher levels of these bioactive lipids in MD subjects than the levels found in *Coriolus* administered MD patients are reported in (**a**–**c**).

**Table 1 ijms-21-00284-t001:** Profile of Mood States (POMS).

	Pre-Therapy (T0)		Post-Therapy (T1)	
	Score		Score	
Group	A	B	A	B
**Anger (0–48)**	28	29	22	29
**Confusion (0–28)**	17	17	10	16
**Depression (0–60)**	41	39	25	37
**Fatigue (0–28)**	16	19	10	19
**Tension (0–36)**	31	29	13	28
**Vigor (0–32)**	19	17	19	16
**Total Mood Disturbance (** **−** **32 to 200)**	114 ± 9.8	116 ± 8.6	61 ± 6.11	113 ± 8.1

**Table 2 ijms-21-00284-t002:** Crisis frequency.

T0	Group A	Group B
**Vertigo Attack Frequency**		
<2 crisis/year	4 (18.1%)	3 (16.6%)
From 3 to 5 crisis/year	10 (45.4%)	8 (44.4%)
From 6 to 8 crisis/year	8 (36.3%)	7 (38.8%)
**Crisis Duration**		
<1 h	4 (18.1%)	4 (22.2%)
From 1 to 7 h	12 (54.5%)	9 (50%)
>24 h	6 (27.2%)	5 (27.7%)
**Duration of Symptoms**		
A few days	15 (68.1%)	11 (61.1%)
Some weeks	6 (27.2%)	6 (33.3%)
A month	1 (4.5%)	1 (5.5%)

**Table 3 ijms-21-00284-t003:** Tinnitus Handicap Inventory (THI).

	Tinnitus Handicap	Inventory	
Pre-Therapy Score	(T0)	Post-Therapy Score	(T1)
Group A	Group B	Group A	Group B
74 ± 2.46	78 ± 2.73	52 ± 1.73 *	74 ± 2.65

* significantly different vs. control untreated MD patients (*p* < 0.05).

**Table 4 ijms-21-00284-t004:** Plasma and lymphocyte content of total, reduced (GSH) and oxidized (GSSG) glutathione in control and MD patients treated with Coriolus.

	Plasma (nmol/mL)			Lymphocyte (nmol/mg Protein)		
	Control	MD	MD + Coriolus	Control	MD	MD + Coriolus
**Total GSH**	16.7 ± 2.1	8.33 ± 3.0 *	14.23 ± 2.4 **	9.81 ± 0.8	5.3 ± 0.7 *	7.3 ± 0.5 **
**GSH**	15.62 ± 2.0	8.44 ± 1.7 *	13.44 ± 1.7 **	9.58 ± 0.6	4.27 ± 0.4 *	7.20 ± 0.5 **
**GSSG**	0.138 ± 0.01	0.169 ± 0.01 *	0.146 ± 0.01 **	0.093 ± 0.01	0.118 ± 0.01 **	0.096 ± 0.006 **
**Ratio GSH/GSSG**	113.2 ± 11	56.9 ± 15*	92.05 ± 13 **	96.5 ± 10	42.6 ± 7.9 *	75.0 ± 9.6 **

***** Significantly different from control (*p* < 0.05). ****** Significantly different from MD alone (*p* < 0.05).

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
