# Peer review of "Nutritional Mushroom Treatment in Meniere’s Disease with Coriolus versicolor: A Rationale for Therapeutic Intervention in Neuroinflammation and Antineurodegeneration"

_ijms, 2019, doi:10.3390/ijms21010284_

Round 1
Reviewer 1 Report
In this article, Scuto et al. assess the protective effect of Coriolus versicolor (CV) mushroom preparation in Meniere’s disease. Specifically, the authors conducted tests to assess profile of mood states and sensorineural hearing loss and showed that it is improved in patients with Meniere’s disease, who received CV biomass preparation for 2 months. Additionally, the authors show that anti-oxidant enzyme and chaperone levels are increased in plasma and lymphocytes from patients treated with CV. Furthermore, indicators of oxidative stress were reduced in patients treated with CV.
The strength of the study lies in use of patient samples to test behavioral and molecular changes. I have following minor comments;
The authors should consolidate some of the figures into one big figure. Results from figures 1-11 can be consolidated into 2 or 3 figures. For figure 11a and b, do the authors conduct more than one replicate?
Author Response
We thank you the reviewer for his positive consideration. Accordingly we have grouped figures for consolidation. Yes we conducted triplicated analyses
Reviewer 2 Report
The results are clear- there are typographical errors that need to be corrected and some abbreviations used without being clarified.
Author Response
We thank you very much the reviewer for his consideration and accordingly we have corrected typographical errors and some abbreviations used clarified.